# Resource-aware construct design in mammalian cells

Roberto Di Blasi [1,2], Mara Pisani[3,4], Fabiana Tedeschi[3,5], Masue M. Marbiah [1,2], Karen Polizzi [1,2], Simone Furini[6], Velia Siciliano [3] & Francesca Ceroni [1,2] ✉

Resource competition can be the cause of unintended coupling between co-expressed genetic constructs. Here we report the quantification of the resource load imposed by different mammalian genetic components and identify construct designs with increased performance and reduced resource footprint. We use these to generate improved synthetic circuits and optimise the co-expression of transfected cassettes, shedding light on how this can be useful for bioproduction and biotherapeutic applications. This work provides the scientific community with a framework to consider resource demand when designing mammalian constructs to achieve robust and optimised gene expression.

Resource competition arises from the simultaneous use of cellular gene expression resources (i.e. polymerases, ribosomes, and other cellular factors) by different expression cassettes, leading to a non-intuitive indirect coupling of their output levels and behaviour[1–3]. In bacteria, the phenomenon has been extensively assessed and quantified[4–6], and many experimental and computational strategies towards its mitigation have been proposed[7–13]. Among these, identification of genetic designs with lower impact on the host resources was shown to be feasible and desirable for optimising synthetic construct performance[5,14].

Although robust heterologous expression is central to mammalian cell engineering and its applications, resource competition in eukaryotes has only recently started to be investigated. Evidence confirmed that gene expression resources are as limiting in mammalian cells as they are in bacteria and that unobvious coupling of genetic circuits can arise due to synthetic cassettes drawing from the same pool of resources[15–19]. However, accurate characterisation of the demand that different construct components place on cellular resources is still missing in mammalian cells, limiting our ability to rationally design genetic constructs with minimised resource footprint towards robust and predictable engineering of these organisms. Understanding resource competition and its impact on construct performance is particularly important when co-expression of different cassettes is desired, something that is often sought after when

engineering mammalian systems, from the use of transfection controls in fundamental research to the expression of multi-gene constructs for bioproduction and biotherapeutic applications[17,20,21].

## Results

### A framework to quantify the resource load of genetic parts

To advance resource-aware construct design in mammalian cells, we developed a framework for quantification of the load imposed by different designs on cellular resources.

To visualise and quantify intracellular resource usage, a test plasmid is co-transfected along with a fluorescence-based capacity monitor, a CMVp-controlled cassette for mKATE expression[15]. When co-expressed in the cell, the monitor and the test plasmid rely on the same pool of cellular resources for gene expression. As the test plasmid uses more resources, less are available for the expression of the capacity monitor, leading to a decrease in its expression levels. Since the monitor cassette lacks regulation, changes in its expression levels can be used as a proxy measure of the available cellular resources (Fig. 1a). We designed a modular test plasmid architecture to enable easy substitution of genetic parts of interest (Fig. S1). This was used to generate a library of synthetic constructs for constitutive and inducible expression that were individually co-transfected with the capacity monitor in HEK293T and CHO-K1 to identify genetic designs with maximised expression and minimised resource load (Fig. 1b).

[1]Department of Chemical Engineering, Imperial College London, South Kensington Campus, London, UK. [2]Imperial College Centre for Synthetic Biology, South Kensington Campus, London, UK. [3]Synthetic and Systems Biology lab for Biomedicine, Instituto Italiano di Tecnologia-IIT, Largo Barsanti e Matteucci, Naples, Italy. [4]Open University affiliated centre, Milton Keynes, UK. [5]University of Naples Federico II, Naples, Italy. [6]Department of Electrical, Electronic and Information Engineering "Guglielmo Marconi", University of Bologna, Cesena, Italy. ✉e-mail: f.ceroni@imperial.ac.uk

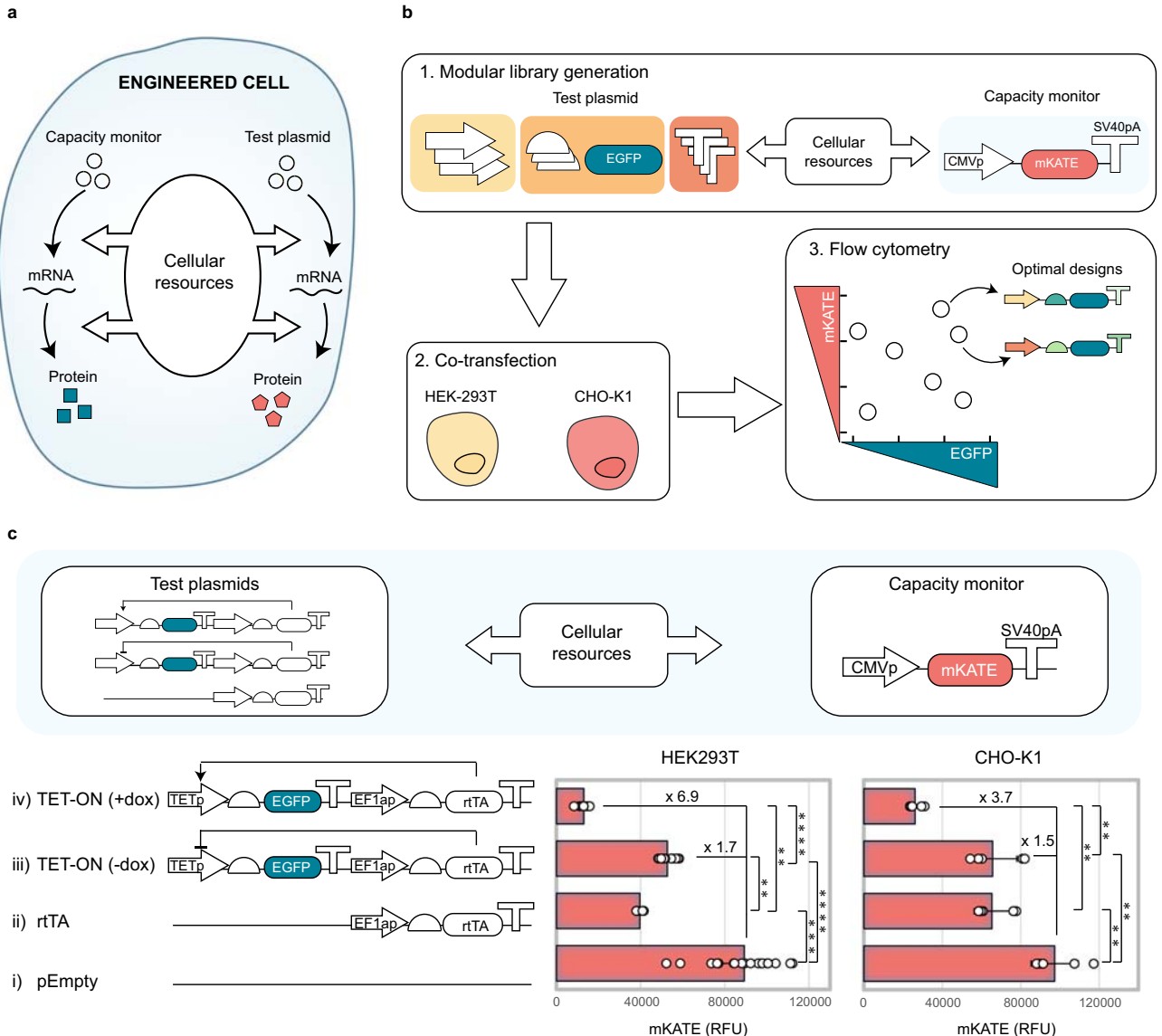

**Fig. 1 | Resource load imposed by synthetic constructs can be measured by co-transfection with a transient capacity monitor. a** A transiently expressed fluorescence-based capacity monitor can be used to measure the usage of gene expression resources by a competing co-transfected test plasmid in mammalian cells. **b** Modular library of synthetic constructs for transient EGFP production developed by varying different construct constituents (promoters, Kozaks and polyAs). Individual plasmids were co-transfected with a CMV-mKATE capacity monitor in HEK293T and CHO-K1. Intracellular fluorescence was measured 48 h post-transfection using flow cytometry. **c** Response of the capacity monitor to resource loading by synthetic designs with increasing complexity in HEK293T and CHO-K1. An empty plasmid (i), a construct constitutively expressing the rtTA transactivator (ii) and the TET-ON inducible system without (iii) and with (iv) dox addition (1 ng/µl) were considered. Capacity monitor expression is reported as mKATE mean RFU ± standard deviation. Number of biological repeats for each sample is reported in Supplementary data file 3. Two sided Mann–Whitney test *P*-value: ****<0.0001, ***<0.0005, **<0.005, *<0.05. Exact *P*-values can be found in Supplementary data file 4. Data analysis is described in the methods section and Supplementary Note 1. Source data are provided in the Source Data file.

We first investigated the sensitivity of the capacity monitor to resource loading by co-transfecting synthetic designs with diverse compositions (Fig. 1c). Specifically, we compared the levels of the capacity monitor in competition with an empty vector (pEmpty) as control of minimal resource demand (i), a constitutively expressed tetracycline transactivator (rtTA) alone (ii), or in combination with its cognate promoter (TETp) driving EGFP expression in the absence (iii) or presence (iv) of the small-molecule inducer doxycycline (dox)[22]. In both HEK293T and CHO-K1, capacity monitor expression decreased as the complexity of the competing test plasmid increased (Fig. 1c, Fig. S2). While co-transfection of the rtTA-expressing plasmid and the uninduced TET-ON-controlled EGFP resulted in a decrease in capacity monitor expression of the same order of magnitude, a further reduction is observed upon dox addition (Fig. 1c). This is consistent with the

expression of two transcriptional units, as opposed to only one in the less complex designs. Reducing the dosage of transfected plasmids did not rescue resource competition (Figure S3a), and induction of a single-copy integrated gene led to resource loading on the capacity monitor (Figure S3b), indicating that resource competition is not limited to co-transfected systems.

We then investigated the role of different genetic elements in driving resource competition. We used our modular backbone to generate a library of EGFP-expressing synthetic designs bearing different promoters, Kozaks, and polyA configurations. We selected seven constitutive promoters (pJB42CAT5, PGKp, hACTBp, SV40p, UBp, CMVp, EF1ap) and one inducible promoter (TETp), three Kozak sequences with different translational efficiency, and six polyA sequences (SV40pA, SV40pA in reverse orientation−SV40pA_rv-,

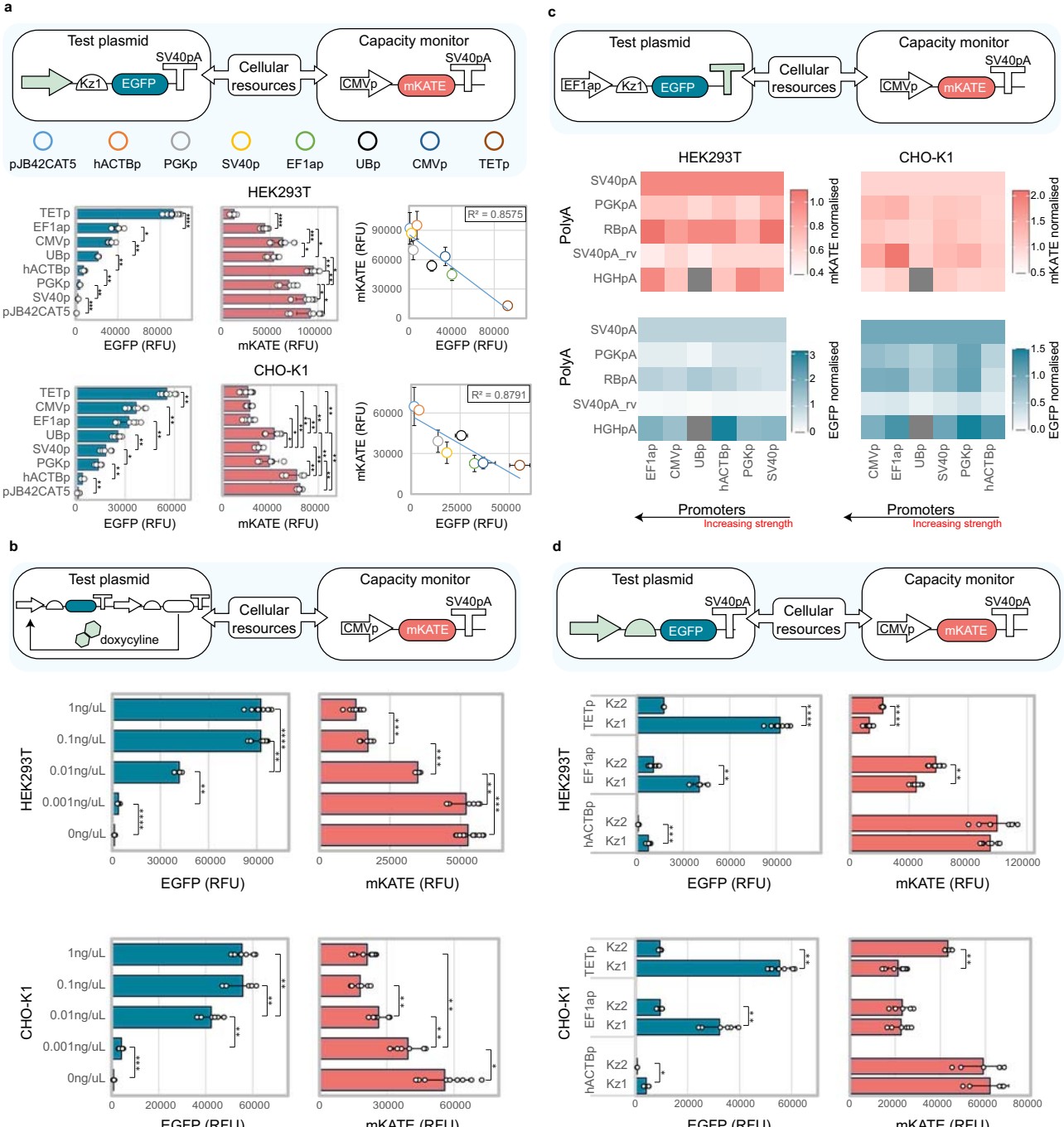

**Fig. 2 | Design features impacting the resource load of synthetic constructs.**
**a** Resource load of seven constitutive promoters and one inducible promoter (TETp, induced with 1 ng/μl). Increased promoter strength of the test plasmid leads to decreased capacity monitor levels in HEK293T (top) and CHO-K1 (bottom).
**b** Resource load of the TET-ON system when induced with increasing dox concentrations in HEK293T (top) and CHO-K1 (bottom). **c** Resource load of different promoter-polyA combinations in HEK293T (left) and CHO-K1 (right). EGFP and mKATE mean RFU values for each pair are normalised to the construct bearing the same promoter in combination with SV40pA. The grey square indicates the UBp-

HGHpA combination, which did not clone correctly and was left out of the analysis.
**d** Resource load of different promoter-Kozak configurations in HEK293T (top) and CHO-K1 (bottom). EGFP output (left) and mKATE monitor levels (right) are shown for hACTB, EF1a and TET promoters. Values are reported as mean RFU ± std. Number of biological repeats for each sample is reported in Supplementary data file 3. Two sided Mann–Whitney test *P*-value: ****<0.0001, ***<0.0005, **<0.005, *<0.05. Exact *P*-values can be found in Supplementary data file 4. Data were analysed as described in the Methods section and Supplementary Note 1. Source data are provided in the Source Data file.

PGKpA, BGHpA, RBpA, HGHpA) (Supplementary data file 1), reasoning they would likely represent critical parameters that influence the resource demand that synthetic constructs place on cellular gene expression resources at the transcriptional and translational levels.

First, we tested the effect that promoter strength imposes on cellular resources. We individually co-transfected a library of test plasmids bearing one of eight promoters along with the capacity monitor. Although, as previously reported[23], equal promoters often display different outputs when adopted in different cell lines,

promoter strength negatively correlated with monitor expression levels in HEK293T and CHO-K1 (Fig. 2a).

This suggests that promoter strength is a crucial parameter in defining the resource footprint of a synthetic construct, with stronger promoters requiring more resources than weaker ones. A few promoters showed better performance than others (i.e. UBp in CHO-K1), where "performance" is defined as the maximisation of both the test plasmid and the capacity monitor expression (Fig. 2a). Similar to what observed for constitutive expression, when the TET-ON system was induced with increasing dox concentrations, increased test plasmid production and corresponding decreased capacity monitor expression levels were observed (Fig. 2b).

To show that our findings are not dependent on the capacity monitor design, we tested different capacity monitors in which the mKATE is controlled by different promoters, namely SV40p, UBp, and EF1ap (Fig. S4). When tested with the promoter library, the additional monitor designs displayed similar behaviour to the original CMV-mKATE capacity monitor, with increasing transcriptional rate from the test plasmid leading to decreased capacity monitor expression. In addition, promoter and cell line-specific effects were observed. These are discussed in Supplementary Note 2.

PolyAs play an essential role in the regulation of eukaryotic gene expression, as they mediate transcriptional termination but also impact mRNA stability, localisation and translation[24,25]. When six different polyAs (SV40pA, SV40pA_rv, PGKpA, BGHpA, RBpA, HGHpA) were assembled within the modular library, diverse impact on the test plasmid output was observed (Fig. 2c, Fig. S5). This is in line with recent work highlighting polyA-dependent tuning of gene expression in mammalian cells[26,27]. Although some polyA sequences consistently resulted in increased or suppressed test plasmid outputs, we observed variability in combination with different promoter sequences (Fig. 2c, Fig. S5)[28]. We also individuated specific polyAs which consistently resulted in suppressed expression levels of the competing capacity monitor (i.e. PGKpA, SV40pA_rv in HEK293T, and SV40pA, HGHpA in CHO-K1) and observed that polyA-based interference on the capacity monitor expression is more dramatic in combination with stronger promoters (especially in CHO-K1) (Fig. 2c). It is worth noting that the effect of equal promoter-polyA combinations on the expression of the competing monitor is cell line-dependent (Fig. S5). To investigate this further, we designed and cloned four additional capacity monitors bearing different polyAs (i.e. SV40pA_rv, HGHpA, PGKpA and RBpA). By conducting competition experiments between the new monitor designs and five shortlisted test plasmids bearing different polyA configurations, we identified distinct polyA-specific competition profiles in addition to the previously observed cell line-specific effects (Fig. S6, Supplementary Note 3).

Although both promoter and polyA selection impact the expression of the competing capacity monitor, whether they do this directly by recruiting transcriptional resources or indirectly by impacting downstream translational resources requires further investigation.

Kozak sequences are directly linked to translational resources as they represent the translation start sites in eukaryotes. Kozak motifs with different translational efficiency have been designed and reported[29]. We thus adopted three previously reported Kozak sequences[30] and tested their behaviour when assembled downstream of different promoters (Fig. S7). We observed that Kz1 and Kz3 displayed similar output for most promoter combinations, whereas Kz2 led to lower test plasmid expression (Fig. S7). When comparing different promoter-Kozak combinations, most of the variation in the capacity monitor expression was explainable by the change in promoter, and significant changes in the monitor expression due to Kozak variation were primarily observed in combination with strong promoters (Fig. 2d, Fig. S7). This suggests that transcriptional resources might be more limiting than translational resources in the tested cell lines, and that the latter can accommodate a higher demand from

synthetic constructs before being saturated. To investigate this further, we cloned three hairpins of increasing minimum free energy (MFE) (Fig. S8a). The hairpins impede translation by sequestering the Kozak sequence. While increasing MFE led to increased suppression of EGFP translation, we observed minimal impact on the capacity monitor expression (Fig S8a), in accordance with our previous data. Interestingly, when similar changes in EGFP expression were obtained by a change in promoter sequence, the capacity monitor expression was dramatically altered (Fig S8a, red asterisks). Finally, we conducted mRNA co-transfection experiments to disentangle the contribution of transcriptional and translational resources to the resource competition landscape. Specifically, we generated two test mRNAs bearing Kozak sequences with different translational efficiency (Kz1-EGFP and Kz2-EGFP) and one monitor mRNA (Kz1-mKATE). Upon individual co-transfection of each test mRNA with the monitor mRNA, we observed no alteration in the mKATE expression levels despite significant differences in EGFP output (Fig. S8b). Altogether, these results are in striking contrast with those previously reported in bacteria, where translational resources play a major role in shaping resource competition[5,8,31–34].

## Identification of designs with minimised footprint

Combining all the library designs can clarify the overall competition landscape and inform on the presence of more efficient design options (Fig. 3a, Fig. S9). Although we observed that greater output from the test plasmids was generally matched by decreased expression of the capacity monitor, some designs appeared more performant than others, enabling the identification of regions where co-expression of both cassettes can be optimised (Fig. 3a, shaded area). To assess how various genetic components affect a construct's resource footprint, we conducted a pairwise comparison of constructs differing in one of the three design elements considered in this study (promoters, Kozaks, or polyA tails) (Fig. 3b). We measured the differences in EGFP and mKATE (ΔEGFP and ΔmKATE) between each pair of constructs bearing (i) different promoters, but identical Kozaks and polyAs; (ii) different Kozaks, but identical promoters and polyAs; and (iii) different polyAs, but identical promoters and Kozaks. These pairs were then grouped into quartiles based on their ΔEGFP, revealing how similar changes in EGFP expression have a different impact on mKATE depending on whether the modification is made to the promoter, Kozak or polyA sequences (Fig. 3b–d). In both cell lines, the promoter is the genetic component with the greatest impact on the capacity monitor expression. Kozak efficiency minorly affects a construct's resource footprint, and only for high ΔEGFP values. This is consistent with our prior observations indicating that translational resources play a minor role in driving resource competition. The role of polyAs is cell-line dependent, and while in HEK293T, increasing ΔEGFP due to polyA variation leads to increased monitor expression, in CHO-K1, polyA-based interference displays the opposite behaviour. We noted that while changing each of the tested genetic parts can lead to the same gain in test plasmid expression, promoters do so by impacting the capacity monitor to a greater extent than Kozaks and polyAs (Fig. 3b–d).

Multi-gene constructs are often needed for the production of biotherapeutic proteins. Different configurations are adopted for the co-expression of two or more genes, such as multiple transcriptional unit arrangements or bi-cistronic modules that use internal ribosome entry sites (IRES) or 2A peptides[35].

Starting from the observation that transcriptional resources are more easily saturated than translational resources in mammalian cells, we compared different design configurations for multi-gene expression to identify designs with lower resource footprints.

We assembled a small library of constructs for the co-expression of EGFP and EBFP either from two separate transcriptional units (TUs) or from one TU using 2A peptides or an IRES sequence (Fig. 3e, f). In

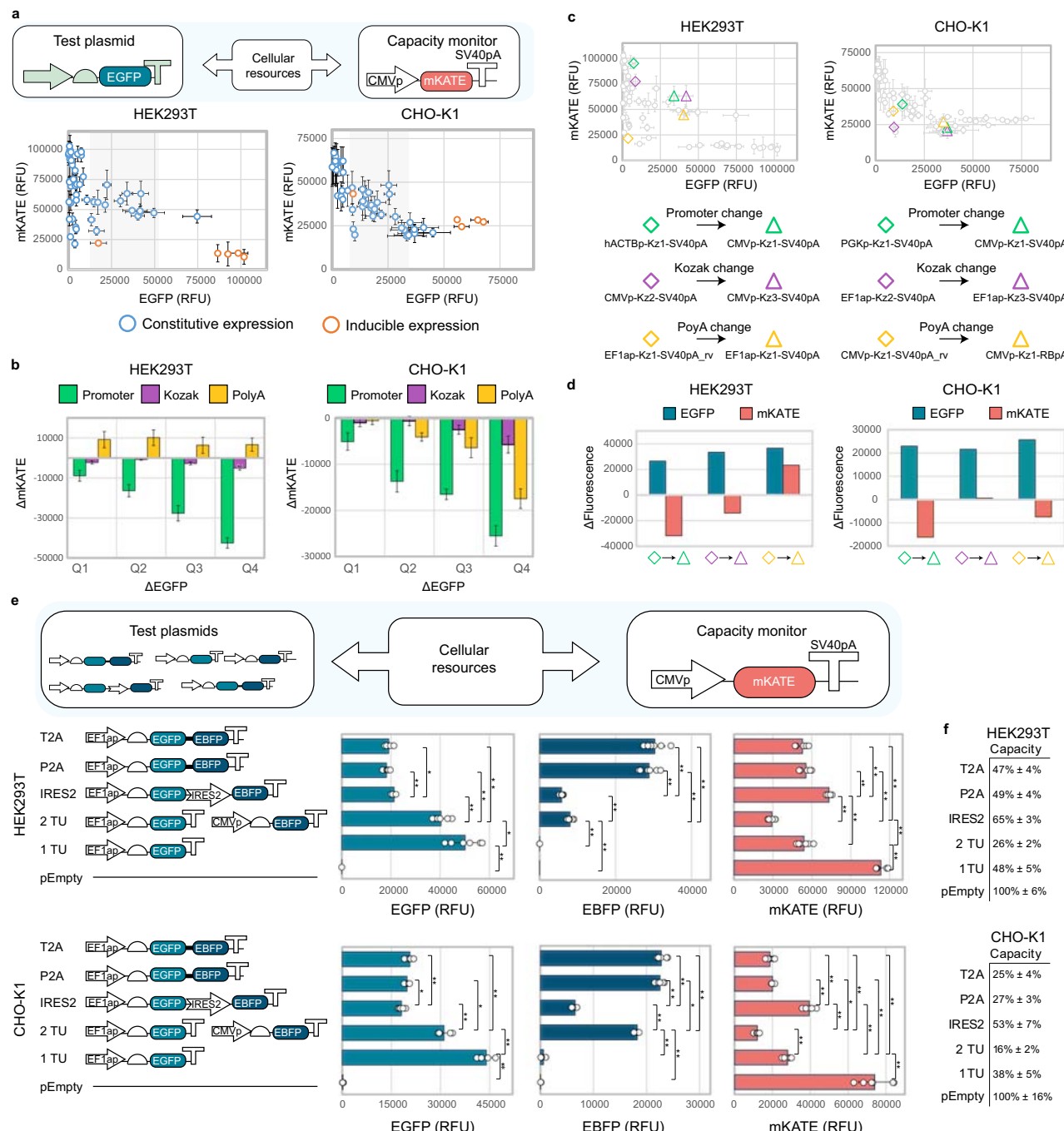

**Fig. 3 | Resource-aware construct design identifies alternatives with maximised performance and minimised resource load. a** Full construct library in HEK293T (top) and CHO-K1 (bottom). Shaded areas identify regions where the co-expression of both cassettes can be optimised. **b** Contribution of promoter, Kozak and polyA to resource load. Vertical bars show the average change in mKATE in each EGFP quartile due to changing promoter, Kozak, or polyA. mKATE and EGFP variation due to promoter change was obtained by comparing all pairs of circuits with identical Kozak and polyA but different promoter sequences. The same method was used to define the list of mKATE and EGFP variation values due to Kozak and polyA. **c** Similar variations in EGFP output achieved by changing promoter, Kozak or polyA result in different impacts on the capacity monitor levels. **d** ΔFluorescence for the construct pairs in Fig. 3c. **e** Test plasmid (EGFP and EBFP) and capacity monitor (mKATE) outputs from single and multi-gene constructs with different designs are shown. The constructs express EGFP (i), EGFP and EBFP from two transcriptional units (ii), EGFP and EBFP linked by IRES (iii) or 2A peptides (iv–v). **f** Remaining cellular capacity for the constructs in Fig. 3e. Test plasmid and capacity monitor expression are reported as mean RFU ± std. Number of biological repeats for each sample is reported in Supplementary Data 3. Two sided Mann–Whitney test $P$-value: ****<0.0001, ***<0.0005, **<0.005, *<0.05. Exact $P$-values can be found in Supplementary data file 4. Data were analysed as described in the Methods section and Supplementary Note 1. Source data are provided in the Source Data file.

both HEK293T and CHO-K1, a stepwise decrease in the monitor expression was observed upon competition with an increasing number of TUs, with the pEmpty retaining higher capacity than a construct expressing one TU and two TUs, with 100%, 48% and 26% of remaining capacity in HEK293T, respectively (Fig. 3e, f). When EBFP and EGFP are arranged in a bi-cistronic configuration by use of IRES and 2A peptides, capacity is rescued to levels comparable to or higher than the expression of a single EGFP-expressing TU (i.e. ≥48% in HEK293T), despite the latter producing only one fluorescent protein (Fig. 3e, f, Fig. S10).

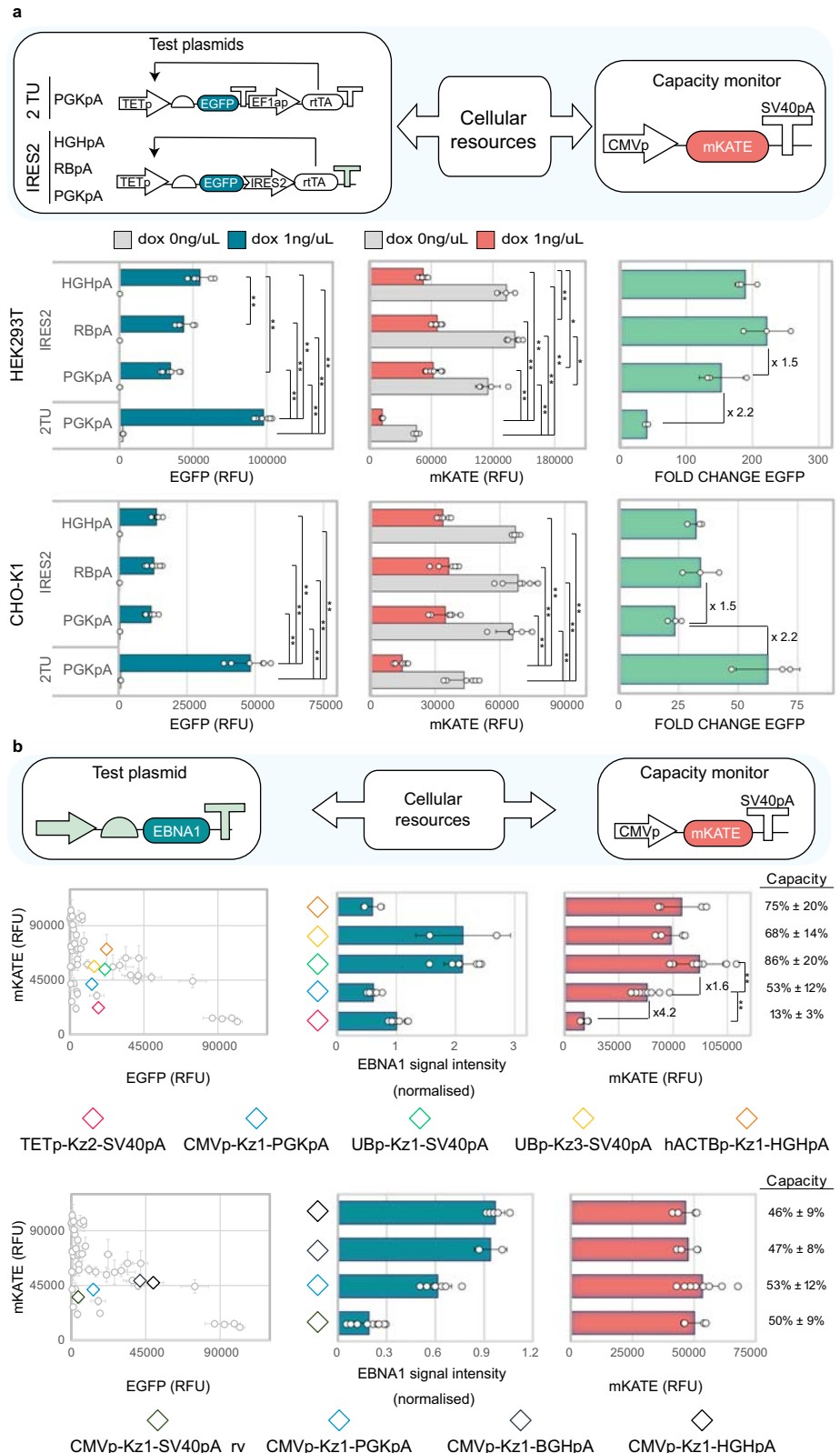

This reinforces our findings, as reducing the number of promoters in the circuit frees up transcriptional resources and dramatically reduces the resource footprint of the multi-gene configuration. However, for both cell lines, although the use of the IRES2 allows higher capacity than 2A peptides, this comes at the cost of reduced output levels of the coding sequence in second position (Fig. 3e, f, S10). This is in line with previous work demonstrating that IRES-dependent

translation is lower than CAP-dependent translation, thus selection of appropriate bi-cistronic configurations (IRES vs 2A peptides) for specific applications may not always line up with the alternative with the lowest resource footprint.

Changing gene syntax on the test plasmid also resulted in different resource uptake (Fig. S11). Placing two TUs on the same plasmid yielded higher monitor expression than identical TUs split between

**Fig. 4 | Resource-aware construct design is useful for optimising genetic circuit performance and co-expression systems. a** Minimisation of the resource load of the TET-ON inducible system by use of an IRES2 sequence. EGFP (left), mKATE (middle) and fold change (ratio of EGFP in the presence and absence of doxycycline induction) (right) are shown. **b** Resource-aware design of EBNA1-expressing constructs in HEK293T. Four designs with different resource footprints and similar EGFP expression are selected (diamonds, top left panel). EBNA1 (top middle panel) and capacity monitor (top right panel) expression levels for each design are shown. Four designs with different EGFP expression levels and similar resource footprints

are selected (diamonds, bottom left panel). EBNA1 (top middle panel) and capacity monitor (top right panel) expression levels for each design are shown. EGFP and capacity monitor expression are reported as mean RFU ± std. EBNA1 expression is reported as normalised signal intensity and was calculated as described in the Methods section. Number of biological repeats for each sample is reported in Supplementary data file 3. Two sided Mann–Whitney test P-value: ****<0.0001, ***<0.0005, **<0.005, *<0.05. Exact P-values can be found in Supplementary data file 4. Data were analysed as described in the Methods section and Supplementary Note 1. Source data are provided in the Source Data file.

two separate plasmids. This was mirrored by the decreased output of one of the two TUs in the linked arrangement compared to the split one (Fig. S11). Interference between adjacent transcriptional units is a well-reported phenomenon[36], but effects on a competing gene were never observed before.

## Optimisation of a genetic switch and co-expression systems
Our results can prove highly relevant to optimise the co-expression of different cassettes. To prove this, we chose two examples demonstrating that (i) it is possible to decrease the resource load of a gene switch widely adopted for gene expression control and that (ii) best-performing designs hold when adopted on a protein of interest for bioproduction and gene therapy applications.

We started by re-considering the TET-ON doxycycline-responsive system adopted earlier in this study, given that this system is widely used in mammalian cell engineering for controlled gene expression (i.e. miRNAs, biotherapeutics, reporters for foundational studies). The system is based on two TUs, one expressing the rtTA transactivator, which, in the presence of dox, can bind and activate an inducible TETp promoter driving the expression of a gene of interest (i.e. an EGFP in this case). In our experiments, the TET-ON system imposed a considerable load on cellular resources (Fig. 1c). We speculate that this is due to the expression of two transcriptional units needed for its functioning.

We designed a new configuration where transcription of both the rtTA and the EGFP is initiated from a single TETp through an IRES or 2A peptide and terminated by different polyAs selected to yield different test plasmid outputs. We observed increased monitor expression levels in both the uninduced and induced conditions when using a single TU configuration compared to the two TUs (Fig. 4a, Fig. S12). Although the resource-efficient designs demonstrated lower EGFP expression in induced conditions, an improvement in fold change (defined as the ratio of EGFP output in the presence and absence of dox) was observed in HEK293T cells. Additionally, we noticed that designs incorporating the more efficient RBpA and HGHpA exhibited higher fold changes than designs carrying PGKpA (Fig. 4a, Fig. S12). Increasing dox and rtTA concentration did not improve EGFP expression for the resource-aware designs (Fig. S13a, b). However, we did observe that the percentage of EGFP-expressing cells was higher when additional rtTA was present for both IRES2 and P2A designs in both cell lines (Fig. S13c). These findings point to the need to find a balance between the desired output and resource footprint, tailored to the specific application.

Our redesigned TET-ON stands as a proof-of-concept that genetic circuits for gene expression control can be designed to impose a lower demand on cellular resources while at the same time displaying improved fold change. Future work could focus on developing circuits with higher maximum output if desired for the application.

Finally, we confirmed that our findings are independent of the type of gene in use. This is highly relevant as biotherapeutic proteins often require the co-expression of multiple cassettes (i.e. antibodies, VLPs, enzymatic complexes), which will inevitably compete for cellular resources. We chose EBNA1, a viral protein often adopted in bioprocessing and gene therapy to improve transient transfection[37–42]. Based on data in Fig. 3a, we designed eight EBNA1-expressing constructs and

co-transfected them with the capacity monitor, working as a proxy for any biotherapeutic of interest. Although we observed no correlation between the EBNA1 and EGFP expression levels of the selected designs, this is in accordance with previous work outlining lack of modularity when the genetic context is changed[43]. However, the expression levels of the capacity monitor did show correlation when in competition with either EGFP or EBNA1 (Fig. S14a-b). This indicates that the capacity monitor output is a good predictor of the resource footprint of a specific design independently of the gene of use. Five of the eight shortlisted designs were selected to yield similar EBNA1 output while displaying different resource footprints (Fig. 4b, top panels). In accordance with our previous characterisation (Fig. 3a–d), designs carrying strong promoters resulted in a higher resource footprint than designs with weaker promoters, with 13% remaining capacity for TETp, 53% for CMVp, and 68-85% for UBp and hACTBp designs (Fig. 4b, top panels). This is in line with the key findings we inferred from Fig. 3a–c. In addition, four designs bearing CMVp-Kz1 and one of four polyA configurations were selected to yield different EBNA1 outputs while maintaining the same resource footprint (Fig. 4b, bottom panels). In line with previous findings (Fig. 2c, Fig. S5), the change of the polyA to more favourable configurations allowed for higher EBNA1 production without increasing the resource footprint.

## Discussion
Robust and predictable heterologous gene expression is critical to foundational and applied biology. Limiting intracellular resources represent a bottleneck to predictable performance when two or more heterologous cassettes are co-expressed in mammalian cells. Here, we present a framework for quantifying the resource load of construct libraries for expression in mammalian cells. By screening a library of synthetic constructs bearing different promoters, Kozaks and polyAs, we show that the design of genetic circuits directly affects their resource footprint. We characterise widely adopted parts for multicistronic expression, such as IRES and 2A peptides, and verify that these parts are crucial in minimising the resource footprint of multigene constructs. We apply our findings to the optimisation of genetic circuit performance (i.e. the TET-ON system), and we demonstrate that our framework remains relevant when applied to other proteins of interest for bioproduction and biotherapy, such as EBNA1. Specifically, we carefully selected genetic parts within the library (i.e. promoters, Kozaks, polyAs) to achieve (i) designs with minimised resource load on similar EBNA1 expression, (ii) designs with increased EBNA1 expression on similar resource load, (iii) designs with increased EBNA1 expression and minimised resource load. Our applications demonstrate the broad applicability of our findings and make this work relevant to anyone working with co-expressed systems in mammalian cells.

Although previous work used a transient capacity monitor to visualise resource competition in mammalian cells[15,16], a direct link between construct design and resource footprint was not demonstrated. Our work gives the scientific community insights into resource allocation in mammalian cells. While earlier work showed that resources are limiting in mammalian cells, contrasting evidence was provided on the role of transcriptional and translational resources[15,16]. By studying circuit components directly linked to transcription or translation, we show that promoters impact resources the most

compared to Kozak sequences. This contrasts with what is reported for bacteria, where translational resources were recognised as the primary source of gene expression burden[5]. Our work also highlights the importance of polyAs in defining the resource landscape of synthetic constructs and points to the need for a deeper understanding of the contribution of different expression resources in different genetic backgrounds. Future work investigating these effects in a wider panel of cell lines could provide a comprehensive analysis of cell line-specific effects. While we report initial evidence that competition arises also between transient and integrated systems, further work will also be needed to expand on this and provide characterisation and analysis of how our findings translate to integrated systems. This will be of invaluable importance considering the wide adoption of stably integrated constructs in mammalian cell engineering and its applications.

Mitigation strategies have been proposed to address the shortcomings caused by resource competition in mammalian cells[15,16,44–47]. One such example is the use of incoherent feed-forward loops (iFFLs). Although iFFLs have been implemented to mitigate unwanted coupling between competing cassettes, they come at the cost of dramatic output suppression of the regulated gene. This limits the implementation of such network motif, as it is unsuitable in settings where output maximisation is sought. By implementing our framework, coupling effects can be mitigated without engineering complex circuits and avoiding suppression of the protein expression levels.

Differently from bacteria, mammalian cells possess compartments and complex regulatory machinery, thus creating a more complex resource allocation scenario than the simple transcription versus translation contribution. Approaches to assess and investigate competition for different types of resources, such as the host-aware framework demonstrated here, provide a way to manage this complexity to enable diverse applications in mammalian synthetic biology.

Ultimately, we provide direct evidence that different synthetic designs can be characterised based on the resource load they impose on mammalian cells. By focusing on genetic parts that make up any functional mammalian gene expression system (i.e. promoters, Kozaks and polyAs), we demonstrate that our framework can be used to optimise co-expression systems, whether targeted towards fundamental or applied biological research.

## Methods

### Bacterial culture and modular cloning
*E. coli* DH5a were used for routine cloning. A modular plasmid architecture was first cloned using the EMMA kit[35] and was then used to generate the construct library. A complete list of plasmids constructed in this study can be found in Supplementary data file 1. Parts and sequences used in this paper can be found in Supplementary data file 2. Primers used in this study can be found in Supplementary data file 5.

### Cell culture
HEK293T (ATCC-CRL-3216) were cultured in DMEM Glutamax (Thermo Fisher 31966047) supplemented with 10% FBS (Life Technologies, 16000-044). CHO-K1 were cultured in MEM α (Sigma Aldrich, M4526) supplemented with 10% FBS, 1% non-essential amino acids (Thermo Fisher, 11140050), and 1% L-glutamine (Thermo Fisher, 25030081). Both cell lines were cultured at 37 °C and 5% CO2.

### DNA transfections
Transfections were performed in a 24-well plate format. Cells were seeded approximately 24 h before transfection in complete media. HEK293T and CHO-K1 were transfected using xTREME GENE HP DNA (Merck, 6366236001) and TransIT-X2 (Mirus, MIR 6003) as transfection reagents, respectively. The number of transfected plasmids was normalised by molar ratio (further information on the transfection protocol can be found in Supplementary data file 3). Cells were detached and prepared for flow cytometry analysis two days after transfection. For rtTA-based induction experiments, unless otherwise specified, 1 ng/µL of doxycycline hyclate (Sigma Aldrich, D9891-25G) was added to the media upon transfection.

### RNA production and transfections
RNAs were transcribed, capped and polyadenylated in vitro using the HiScribe® T7 ARCA mRNA Kit (NEB, E2060S) following the manufacturer's instructions. RNAs were then purified from the in vitro reaction using RNA Clean & Concentrator-25 (Zymo Research, R1017) as per the manufacturer's instructions. The purified RNAs were then co-transfected in HEK293T using TransIT®-mRNA Transfection Kit (Mirus, MIR 2225). 0.25ug of test and monitor RNAs (i.e. 0.5ug in total) were transfected per well (further information on transfection protocol can be found in Supplementary data file 3). Cells were detached using Trypsin (Thermo Fisher, 15400054) and prepared for flow cytometry analysis two days after transfection.

### Flow cytometry and data analysis
Transfected cells were detached, centrifuged, resuspended in DPBS (Thermo Fisher 14190250), and filtered to disrupt any cell clumps. HEK293T and CHO-K1 were analysed with the Attune NxT flow cytometer (Thermo Fisher). For each sample, 10000 viable cells were recorded. Transfected cells were selected using an OR gate approach with fuzzy logic (see Supplementary Note 1). All data points lower than Q1-1.5*IQR and higher than Q3 + 1.5*IQR- where IQR stands for the interquartile range—were considered outliers and excluded from further analysis. Mann–Whitney test was used to assess statistical significance. The code used for data analysis is freely available at https://github.com/sfurini/gating_ResourceAwareDesign.git. Capacity was calculated as a percentage of the total capacity monitor expression measured upon co-transfection of the pEmpty plasmid (i.e. in the absence of competition). Error is expressed as ± % and is calculated from standard deviation values.

### Statistics and reproducibility
No statistical method was used to predetermine sample size. The Investigators were not blinded to allocation during experiments and outcome assessment. All data points lower than Q1-1.5*IQR and higher than Q3-1.5*IQR-where IQR stands for interquartile range-where considered outliers and excluded from further analysis. Number of repeats for each sample are reported in Supplementary data file 3. Exact P-values are reported in Supplementary data file 4.

### Western Blot
Transfected HEK293T cells were detached, centrifuged, and washed in ice-cold DPBS. Cell pellets were lysed in RIPA buffer (Sigma) containing 1% v/v protease inhibitor cocktail (Sigma). The protein concentration of clarified lysates was determined using a BCA assay (Thermo Fisher). Lysates were denatured, and 20 µg were loaded into SDS PAGE gels. Proteins were transferred onto nitrocellulose membranes (Thermo Fisher) and probed for EBNA1 (Merck, clone 1H4, monoclonal, cat MABF2800, dilution 1:1000) and a loading control, Vinculin (Sigma, clone hVIN-1, monoclonal, cat V9264, dilution 1:5000). Secondary antibodies (Life Technologies, Goat anti-Rat IgG A18868, polyclonal, and Abcam Rabbit Anti-Mouse IgG H&L (Alkaline Phosphatase) ab6729, dilutions 1:10000) were detected using the Novex™ AP Chromogenic Substrate (Invitrogen). Image J[48] was used for densitometry analysis. To obtain a normalisation factor specific to each lane, the Vinculin signal for each lane was normalised against the lane with the strongest signal. This was then multiplied by the EBNA1 signal of the same lane to get a normalised EBNA1 experimental signal. Finally, to control for inter-blot variability, normalised EBNA1 values were divided once more against the average normalised signal of each lane present within the blot (except the negative control).

## Reporting summary

Further information on research design is available in the Nature Portfolio Reporting Summary linked to this article.

## Data availability

All source data for the data sets and figures presented in the manuscript are available at "Roberto, Di Blasi; Mara, Pisani; Tedeschi, Fabiana; Marbiah, Masue; Polizzi, Karen Marie; Furini, Simone; et al. (2023): Resource-aware construct design in mammalian cells. Figshare. Dataset. https://doi.org/10.6084/m9.figshare.22659211" and the provided Source Data file. Constructs are available upon request to the corresponding author. Plasmid maps are available in the source data. Source data are provided with this paper.

## Code availability

The code used for data analysis is freely available at https://github.com/sfurini/gating_ResourceAwareDesign.git and https://doi.org/10.5281/zenodo.7956982.

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

## Acknowledgements

The authors thank Tom Ellis and Cleo Kontoravdi for their support, thoughts and advice during the project. R.D.B. was supported by the Imperial College Chemical Engineering PhD scholarship. M.M.M. and K.P. were supported by the Biotechnology and Biological Sciences Research Council (grant BB/S006206/1). V.S. was supported by the ERC Starting Grant Synthetic-TrEX #852012.

## Author contributions

F.C., V.S. and R.D.B. designed the research; R.D.B., M.P., F.T. and M.M.M. performed experiments; R.D.B., S.F. and F.C. analysed the data; F.C., V.S., K.P. contributed funding; F.C., R.D.B. and S.F. wrote the paper; all authors read and edited the manuscript.

## Competing interests
The authors declare no competing interests.
