## [Peer Review File · Nature Communications]

Reviewers' Comments:

Reviewer #1:

Remarks to the Author:

Roberto et al. present a framework for quantifying the resource load of genetic components/circuits in mammalian cells using a co-transfected "capacity monitor" plasmid (constitutive expressed mKate reporter), extending a concept previously demonstrated in *E. coli* (PMID: 25849635, 29578536). By screening a library of synthetic constructs bearing different promoters, Kozak sequences, poly A sequences, and various configurations of multigene transcription units (TUs), they show that the design of genetic circuits directly affects their resource footprint as quantified by the capacity monitor. Using this framework, the authors arrive at a few interesting findings. First, they find that gene expression burden in mammalian cells is most sensitive to promoter choice, indicating that promoter choice is a key parameter in defining the resource footprint of synthetic circuits and suggesting that cellular gene expression resources are limited at the transcriptional level in mammalian cells (in contrast to in prokaryotic systems). Second, they characterize multi-gene designs and find that bi-cistronic designs (e.g. IRES, 2A) have reduced resource load than equivalent multiple transcriptional unit arrangements. Finally, they apply their capacity monitor to identify resource-aware designs for a TET-ON inducible system and show that their framework is relevant for other proteins of interest (e.g. EBNA1).

Understanding the interaction between synthetic circuits and native cellular machinery is an emerging and important area of synthetic biology. Its application to mammalian cell engineering could inform genetic designs for enhanced bioproduction and biotherapeutics. While conceptually not novel, the capacity monitor is a simple and potentially powerful tool, allowing many circuit designs and genetic combinations to be rapidly screened for their resource footprint. This can be useful for identifying the subset of circuits with minimized resource footprint from a set of circuits with equivalent functional outputs. Overall, the work is impressive in the scope of genetic designs that were tested, and some of the insights and optimized designs extracted should be of interest to the mammalian synthetic biology and adjacent communities. However, I have some key comments and questions for the authors to consider and address before making a decision on the suitability of the manuscript.

1. With respect to the results related to Fig. 2: The authors found that promoter choice drives the greatest variation in the capacity monitor. However, isn't this simply explained by the fact that the promoters drive the largest variation in gene expression output, relative to the Kozak's and polyA's tested? I know this is partly addressed by the authors, but I think this point should be clarified and expanded upon.
2. Based on the results of Fig. 2, the authors suggest that, in mammalian cells, transcriptional resources may be more limiting than translational resources. If so, I think this would be an a very interesting and important finding. To substantiate it, the authors should seek out measurements that can provide more direct evidence of transcriptional resource sharing / depletion.
3. The finding that polyA-based variation in the capacity monitor is cell-line dependent is interesting. Can the authors speculate as to why this may be?
4. With respect to the results of Fig. 3D (multi-gene constructs): the EGFP / EBFP reporter outputs for the various designs tested should really be displayed in the main figure, alongside with the capacity monitor. Or perhaps the authors could display a correlation plot. Again, this returns to the comments above about how much of the capacity monitor variation can be explained by the circuit output – presumably the interesting cases are the ones in which it cannot. Here it would naively seem that the 2 TU design is driving the highest levels of both EGFP and EBFP levels.
5. The TET-ON design exploration is interesting and could be impactful, given how widely used this inducible system is. However, the results and interpretation of the results need to be clarified. For example, it appears that the optimized resource-aware (multi-cistronic) designs, while driving significantly lower induction output levels relative to the canonical 2 TU design, lead to lower leakage which manifests as an improvement in fold change levels, at least in HEK293T cells. This represents an improvement in at least one gene expression dimension (fold change). Given these

tradeoffs, the authors should be clear about what they mean by maximized performance through resource-aware design optimization. Interestingly, this result does not appear to be consistent in CHO-K1 cells. The authors speculate "that the improved system may not reach full induction with the usual dox concentration, and that increased concentrations may be needed to improve the system dynamics in CHO-K1." The authors should really provide evidence for this. Another interpretation is that they are getting lower rTta expressed from the new designs.

6. With respect to the results of Fig. 4b: only three samples are taken to draw correlation between EGFP and EBNA1. In order to draw a reliable correlation, the authors should choose more points to show whether there is reliable coincidence. Moreover, the interpretation that "the higher-than-expected EBNA1 output from the UBp construct shows how dramatic cooptimisation of the output of both competing cassettes can be achieved by careful selection of genetic parts" seems unusual. Wasn't the result out of expectation or not coincident with their framework?

Minor comments and critiques:

1. The authors should clarify what TETp design is used in Fig. 2a. I assume this is a constitutive version of the promoter?
2. Did the author normalize the number of each plasmid when transfecting cells? If so, was it done by molar ratio or by weight? This should be indicated either in the text or in the methods
3. In Fig. 3c, in the configuration of UBp-HGHpA, there is no expression of even in the capacity monitor plasmid. Can the authors explain?
4. In the Capacity Table at the right of Fig. 3d, I recommend changing the listed order to be consistent with the plots on the left.
5. Please clarify what cell line was used for Fig. 4 and Fig. S10-11? HEK293T or CHO-K1?

Reviewer #2:

Remarks to the Author:

In this study, the authors characterized the role of resource competition in crosstalk between different cargos in mammalian cells, especially HEK293 and CHO cells. By testing various promoters and other element combinations, the authors found a negative correlation between two output proteins. However, the finding is obvious, and the underlying mechanism and exploration are not supportive. Such competition has already explored in mammalian synthetic biology area (pmc4364186). While almost all the assays were performed by transfection, each expression cassette's intensity is regulated by promoter intensity and the transfection dose. To reach a balance between multiple cassettes, the intensity would be simply optimized by testing different doses of plasmids. Whether such competition existed in transfection with high plasmid dose is unclear. Additionally, the actual applications that benefit from this research did not show. Overall, a specialized journal would be more suitable for this manuscript.

Major concerns:

1. Almost all the assays were performed by transfection. The intensity of each expression cassette is not only regulated by the promoter intensity but also by the transfection dose. To reach a balance between several plasmids, the dose can be optimized to give rise to more precise control.
2. Since the transfection in HEK293 cells can deliver a much high level of DNA into the cells, which will occupy a large portion of the resource. However, whether this phenomenon would exist is unclear when researchers utilize other delivery approaches, including AAV, Lentivirus, or mRNA delivery. Additionally, whether the competition only existed in transfection with much high level of plasmids is unclear.
3. Authors contributed the underlying mechanism to transcriptional regulation rather than translation regulation due to the apparent relationship found in the promoters chosen but not the Kozak sites. The Kozak sequence is not a clear translational start signal. The strong transcription by powerful promoters such as CMV would affect all the resource allocation. Deep exploration to

test the critical resources, including ribosome, polymerase, transcriptional factors, or tRNA, would be helpful for further research.

4. The authors did not provide applications that face the competition problem. It would be great if the authors could show the applications to reveal their importance.

Point-by-point responses to reviewer comments

Reviewer #1

Remarks to authors: Roberto et al. present a framework for quantifying the resource load of genetic components/circuits in mammalian cells using a co-transfected “capacity monitor” plasmid (constitutive expressed mKate reporter), extending a concept previously demonstrated in *E. coli* (PMID: 25849635, 29578536). By screening a library of synthetic constructs bearing different promoters, Kozak sequences, poly A sequences, and various configurations of multigene transcription units (TUs), they show that the design of genetic circuits directly affects their resource footprint as quantified by the capacity monitor. Using this framework, the authors arrive at a few interesting findings. First, they find that gene expression burden in mammalian cells is most sensitive to promoter choice, indicating that promoter choice is a key parameter in defining the resource footprint of synthetic circuits and suggesting that cellular gene expression resources are limited at the transcriptional level in mammalian cells (in contrast to in prokaryotic systems). Second, they characterize multi-gene designs and find that bi-cistronic designs (e.g. IRES, 2A) have reduced resource load than equivalent multiple transcriptional unit arrangements. Finally, they apply their capacity monitor to identify resource-aware designs for a TET-ON inducible system and show that their framework is relevant for other proteins of interest (e.g. EBNA1). Understanding the interaction between synthetic circuits and native cellular machinery is an emerging and important area of synthetic biology. Its application to mammalian cell engineering could inform genetic designs for enhanced bioproduction and biotherapeutics. While conceptually not novel, the capacity monitor is a simple and potentially powerful tool, allowing many circuit designs and genetic combinations to be rapidly screened for their resource footprint. This can be useful for identifying the subset of circuits with minimized resource footprint from a set of circuits with equivalent functional outputs.

Overall, the work is impressive in the scope of genetic designs that were tested, and some of the insights and optimized designs extracted should be of interest to the mammalian synthetic biology and adjacent communities.

We thank the reviewer for the positive comments on our manuscript, acknowledging its importance for the field and its interest in informing on resource-aware design considerations.

However, I have some key comments and questions for the authors to consider and address before making a decision on the suitability of the manuscript. 1. With respect to the results related to Fig. 2: The authors found that promoter choice drives the greatest variation in the capacity monitor. However, isn't this simply explained by the fact that the promoters drive the largest variation in gene expression output, relative to the Kozak's and polyA's tested? I know this is partly addressed by the authors, but I think this point should be clarified and expanded upon.

We thank the reviewer for the comment, allowing us to expand some of our considerations on what we believe is a crucial finding of the manuscript. We tried to highlight this concept in the first version of the manuscript. In Fig. 3b, the differences in EGFP and mKATE (Δ EGFP and

Δ mKATE) for promoters were calculated comparing all pairs of constructs with different promoters but identical Kozak and polyA, and consequently, they reflect changes in expression due to the promoter sequence alone. An analogous strategy was used to calculate Δ EGFP and Δ mKATE caused by changing the Kozak or the polyA sequence. Then, these differences in fluorescence were grouped into quartiles based on Δ EGFP (i.e. each quartile contains pairs of samples in which a similar Δ EGFP was obtained by modification of the promoter (green bars), the Kozak sequence (purple bars), or the polyA sequence (yellow bars). Figure 3b provides evidence that similar changes in EGFP expression have a stronger impact on mKATE when the changes are due to the modification of the promoter sequence, as opposed to alterations in the Kozak or polyA.

Fig. 3c shows representative examples of the different impacts of the promoter, Kozak, and polyA. The pairs of diamond/triangle markers with the same colour correspond to pairs of constructs with i) the same Kozak, polyA and different promoter (green markers); ii) the same promoter, polyA and different Kozak (purple markers); iii) the same promoter, Kozak and different polyA (yellow markers). Δ EGFP is almost identical for the three pairs, showing differences in Δ mKATE. These data indicate that for the same variation in EGFP expression, the impact on mKATE levels is more significant if it is the promoter being changed in the construct with respect to the case where the Kozak or the polyA are replaced.

To further address the reviewer's concerns, we produced a new figure panel (Fig. 3d). Here the Δ EGFP and Δ mKATE were plotted for the pairs of constructs highlighted in Fig. 3c. Although the selected pairs of constructs show a comparable Δ EGFP, the effect on mKATE (Δ mKATE) is more pronounced when the promoter is modified compared to the modifications of the Kozak and polyA. This figure hopefully provides better visualisation of the concept and reiterates a key finding that the reviewer highlighted. We provide the updated Fig. 3 below.

To corroborate our claims, we also designed hairpin-containing constructs described in Point 2 below. Both the EF1ap+H2 and hACTBp-Kz1-SV40pA constructs exhibit comparable levels of EGFP output. However, the latter shows significantly higher mKATE expression than the former (please refer to point 2 below). Overall, the change of promoter drives the greatest variation in the expression of a competing capacity monitor compared to the other genetic parts tested in this study.

We have now added a new paragraph in the manuscript to further clarify this data (line 156):

"To assess how various genetic components affect a construct's resource footprint, we conducted a pairwise comparison of constructs differing in one of the three design elements considered in this study (promoters, Kozaks, or polyA tails) (Fig. 3b). We measured the differences in EGFP and mKATE (Δ EGFP and Δ mKATE) between each pair of constructs bearing i) different promoters, but identical Kozak and polyAs; ii) different Kozaks, but identical promoters and polyAs; and iii) different polyAs, but identical promoters and Kozaks. These pairs were then grouped into quartiles based on their Δ EGFP, revealing how similar changes in EGFP expression have a different impact on mKATE depending on whether the modification is made to the promoter, Kozak or polyA sequences (Fig3b-d). In both cell lines, the promoter is the genetic component with the greatest impact on the capacity monitor expression. Kozak

efficiency minorly affects a construct's resource footprint and only for high Δ EGFP values. This is consistent with our prior observations indicating that translational resources have a minor role in driving resource competition."

Fig. 3. Resource-aware construct design identifies alternatives with maximised performance and minimised resource load (a) Full construct library in HEK293T (top) and CHO-K1 (bottom). Shaded areas identify regions where the co-expression of both cassettes can be optimised. (b) Contribution of promoter, Kozak and polyA to resource load. Vertical bars show the average change in mKATE in each EGFP quartile due to changing promoter, Kozak, or polyA. mKATE and EGFP variation due to promoter change was obtained by comparing all pairs of circuits with identical Kozak and polyA but different promoter sequences. The same method was used to define the list of mKATE and EGFP variation values due to Kozak and polyA. (c) Similar variations in EGFP output achieved by changing promoter, Kozak or polyA result in different impacts on the capacity monitor levels. (d) Δ Fluorescence for the construct pairs in Fig. 3c. (e) Test plasmid (EGFP and EBFP) and capacity monitor (mKATE) outputs from single and multi-gene constructs with different designs are shown. The constructs express EGFP i), EGFP and EBFP from two transcriptional units ii), EGFP and EBFP linked by IRES iii) or 2A peptides iv-v). (f) Remaining cellular capacity for the constructs in Fig. 3e. Test plasmid and capacity monitor expression are reported as mean RFU \pm std. Number of biological repeats for each sample is reported in Table S3. Mann-Whitney test P value: **** <0.0001 , *** <0.0005 , ** <0.005 , * <0.05 . Data were analysed as described in the Methods section and Supplementary Note 1.

2. Based on the results of Fig. 2, the authors suggest that, in mammalian cells, transcriptional resources may be more limiting than translational resources. If so, I think this would be an a very interesting and important finding. To substantiate it, the authors should seek out measurements that can provide more direct evidence of transcriptional resource sharing / depletion.

We thank the reviewer for the comment that allows us to add further characterisation to our work and strengthen our results. When deciding the best approach to pinpoint the contribution of transcriptional load in mammalian cells, we excluded qPCR because evidence suggests that the expression of housekeeping genes commonly used in qRT-PCR experiments can vary due to resource competition-Frei *et al.* showed variation in GAPDH mRNA levels under conditions of resource limitation.

To address the reviewer's comment, we cloned three hairpins that work by sequestering the Kozak sequence, impeding translation. We were inspired by our previous research, where a similar approach was used to investigate the contribution of competition for transcriptional and translational resources in bacteria². The figure below shows the structure of the novel constructs we designed. We designed three RNA hairpins with increasing Minimal Free Energy (MFE), as previously described³. These hairpins were cloned at the 5'UTR of EF1ap-Kz1-SV40pA and worked as expected, with hairpins with increasing MFE causing a higher suppression of EGFP translation. Comparing the strongest hairpin (EF1ap + H3), which almost completely represses EGFP expression, with the no-hairpin construct (EF1ap), the rescue in capacity detected by the mKATE monitor is minimal (figure below, now Fig. S8a in the manuscript). However, when the decrease in EGFP output is obtained by a change to a weaker promoter (hACTBp), we observe a marked rescue in the expression of the competing monitor. Notably, the EF1ap + H2 and the hACTBp constructs (highlighted by red asterisks in the figure) express EGFP at a similar level but with a very different effect on the expression of the competing mKATE. These data demonstrate that transcriptional resources are more limiting than translational resources, as the same variation in output causes i) rescue of the competing mKATE expression when transcriptional resources are released (i.e. by changing the strong EF1ap to a weaker hACTBp promoter), ii) minimal rescue on the competing mKATE expression when translational resources are released (i.e. by introducing hairpins in the 5'UTR to block translation). We have now included this figure in the Supplementary Information (Fig. S8a).

To further corroborate our findings, we performed mRNA transfection experiments reasoning this would allow us to isolate the impact on translational resources. We *in-vitro* transcribed two test mRNAs bearing Kozak with different translational efficiency (Kz1-EGFP and Kz2-EGFP) and one monitor mRNA (Kz1-mKATE). All mRNAs have been capped and polyadenylated. The two test mRNAs were individually transfected into HEK293T along with the monitor mRNA. From the figure below, while the EGFP expression from the two different mRNAs is very different, the mKate expression is the same. This experiment stands as further proof of our hypothesis. These data have now been included in the Supplementary Information (Fig. S8b).

Overall, these two additional sets of experiments demonstrate that transcriptional resources are more limiting than translational resources in mammalian cells. Both experiments are coherent with our initial claim, and we hope they fully satisfy the reviewer's request for further evidence.

Fig. S8. Impact of translational efficiency on resource competition profiles. (a) Three hairpins (H1, H2 and H3) were cloned at the 5'-UTR of the EF1ap-Kz1-SV40pA construct. Each hairpin displays a different efficiency in sequestering the Kozak sequence, thus modulating translation. The impact of the designs on the capacity monitor was compared with that of two designs with no hairpin bearing either EF1ap or hACTBp in HEK293T (top panel) and CHO-K1 (bottom panel). Red asterisks identify two constructs with similar EGFP output, one with a stronger promoter and H2, and one with a weaker promoter and no hairpin. **(b)** Two test mRNAs bearing Kozak sequences with different strength (Kz1-EGFP and Kz2-EGFP) and one monitor mRNA (Kz1-mKATE) were generated by *in-vitro* transcription. All mRNAs were capped and polyadenylated. Test mRNAs and monitor mRNA were co-transfected in HEK293T. Expression levels are reported as mean RFU \pm std. Number of biological repeats for each sample is reported in Table S3. Mann-Whitney test P value: ****<0.0001, ***<0.0005, **<0.005, *<0.05. Data were analysed as described in the Methods section and Supplementary Note 1.

We have now added a new paragraph in the manuscript that discusses the new data (line 134):

“To investigate this further, we cloned three hairpins of increasing minimum free energy (MFE) (Fig. S8a). The hairpins impede translation by sequestering the Kozak sequence. While increasing MFE led to increased suppression of EGFP translation, only a minimal impact on the capacity monitor expression was observed (Fig S8a), in accordance with our previous data. Interestingly, when similar changes in EGFP expression were obtained by a change in promoter sequence, the capacity monitor expression was dramatically altered (Fig S8a, red asterisks). Finally, we conducted mRNA co-transfection experiments to disentangle the contribution of transcriptional and translational resources to the resource competition landscape. Specifically, we generated two test mRNAs bearing Kozak sequences with different strength (Kz1-EGFP and Kz2-EGFP) and one monitor mRNA (Kz1-mKATE). Upon individual co-transfection of each test mRNA with the monitor mRNA, we observed no alteration in mKATE expression levels despite significant differences in EGFP output (Fig. S8b).”

3. The finding that polyA-based variation in the capacity monitor is cell-line dependent is interesting. Can the authors speculate as to why this may be?

We thank the reviewer for the comment that allows us to expand on the interesting role of polyAs in shaping resource competition in mammalian cells. To address the reviewer's comment, we have now added Supplementary Note 3 (reported here below), an additional supplementary figure (Fig. S6) and a paragraph to the text referring to these (below).

Supplementary Note 3. PolyA role in shaping resource competition

Our findings indicate that the effect of polyAs on the expression of a competing cassette is cell-line dependent. To further characterise the role of polyAs in shaping resource competition, we cloned different capacity monitor bearing one of four polyAs (i.e. SV40pA_rv, HGHpA, PGKpA and RBpA). We tested the novel monitor designs in competition with a few shortlisted test designs bearing EF1ap, Kz1 and one of the five polyAs previously adopted. Our data (Fig. S6) demonstrate that the trend observed in the monitor expression is cell line- and polyA-dependent. In CHO-K1, we noticed decreasing monitor expression with increasing EGFP due to a change of polyA. In HEK293T, a similar trend was also observed for the monitors bearing PGKpA, RBpA, and HGHpA. However, for the monitors with SV40pA and SV40pA_rv, we observed the opposite behaviour, where increasing EGFP outputs led to an increase in capacity monitor expression.

PolyAs are involved in transcriptional termination, mRNA stability, and translation⁸. Our results suggest a resource bottleneck in at least one of these processes. Accounting for a resource bottleneck explains why polyAs driving a higher EGFP expression leads to decreased expression of the competing capacity monitor. To explain cell-line and polyA specific effects we speculate that the availability of critical cellular factors involved in polyA-mediated biological processes might be crucial in understanding this complexity. An example of cellular factors interacting with polyAs are the cytoplasmic polyA-binding proteins (i.e. PABP), involved in mRNA stability and translation. PABPs bind the polyA, protecting the mRNAs from cellular deacetylation complexes, which trim polyA tails, and increasing mRNA stability⁸. PABPs also bridge the interaction between the polyA tail and eIF4G at the 5'-UTR of the mRNA, promoting mRNA circularisation and enhancing translation, something referred to as the closed-loop translation initiation model⁹. Evidence of cell line-dependent PABP availability and polyA sequence-dependent PABP activity has been previously reported^{10,11}. Our work provides the first evidence for cell line- and sequence-dependent impact of polyAs in shaping resource competition profiles. To pinpoint which specific resources (i.e. the ones involved in transcriptional termination, mRNA stability or translation regulation) are the cause of the observed behaviour will require further experimental characterisation.

References

8 Passmore, L. A. & Collier, J. Roles of mRNA poly(A) tails in regulation of eukaryotic gene expression. *Nat Rev Mol Cell Biol*, doi:10.1038/s41580-021-00417-y (2021)

9 Wang, Z. & Kiledjian, M. The poly(A)-binding protein and an mRNA stability protein jointly regulate an endoribonuclease activity. *Mol Cell Biol* 20, 6334-6341, doi:10.1128/MCB.20.17.6334-6341.2000 (2000)

10 Adivarahan, S. *et al.* Spatial Organization of Single mRNPs at Different Stages of the Gene Expression Pathway. *Mol Cell* 72, 727-738 e725, doi:10.1016/j.molcel.2018.10.010 (2018).
 11 Castellano, L. A. & Bazzini, A. A. Poly(A) tails: longer is not always better. *Nat Struct Mol Biol* 24, 1010-1011, doi:10.1038/nsmb.3509 (2017).

Fig. S6. Sequence- and cell line-dependent polyA-based resource competition. The resource load of test constructs bearing five polyAs downstream of EF1ap-Kz1 was measured in HEK293T (left panels) and CHO-K1 (right panels). The constructs were tested in competition with five different capacity monitors engineered to contain one of five polyA sequences. Test plasmid and capacity monitor expression levels are reported as mean RFU \pm std. Number of biological repeats for each sample is reported in Table S3. Data were analysed as described in the Methods section and Supplementary Note 1.

We have now added a new paragraph in the manuscript referring to the new data (line 114):

“It is worth noting that the effect of the same polyA sequences on the expression of the competing monitor is cell line-dependent (Fig. S5). To investigate this further, we designed and cloned four additional capacity monitors bearing different polyAs (i.e. SV40pA_{rv}, HGHpA,

PGKpA and RBpA). By conducting competition experiments between this library and the polyA test plasmids, we identified distinct polyA-specific competition profiles in addition to the previously observed cell line-specific effects (Fig. S6, Supplementary Note 3)”.

4. With respect to the results of Fig. 3D (multi-gene constructs): the EGFP / EBFP reporter outputs for the various designs tested should really be displayed in the main figure, alongside with the capacity monitor. Or perhaps the authors could display a correlation plot. Again, this returns to the comments above about how much of the capacity monitor variation can be explained by the circuit output – presumably the interesting cases are the ones in which it cannot. Here it would naively seem that the 2 TU design is driving the highest levels of both EGFP and EBFP levels.

We thank the reviewer. As suggested, we have added the EGFP/EBFP outputs to the new Fig. 3e. We have also added a plot to aid visualisation (displayed below), in line with the reviewer’s concern. We have included this figure in the Supplementary Information (Fig. S10).

Fig. S10. Test plasmid and capacity monitor outputs of multi-gene constructs. Test plasmid (EGFP, EGFP) and capacity monitor (mKATE) expression for single and multi-gene designs. EGFP, EBFP, and mKATE expression levels are reported as mean RFU values normalised to the fluorescence value of the corresponding reporter in the 2TU configuration. Number of biological repeats for each sample is reported in Table S3. Data were analysed as described in the Methods section and Supplementary Note 1.

In Fig. S10, EGFP, EBFP, and mKATE expression levels are reported as mean RFU values normalised to the fluorescence value of the corresponding reporter in the 2TU configuration. This figure makes it easier to visualise the expression of the three fluorescent proteins relative to the 2TU configuration. In particular:

- IRES2 construct: the increase in mKATE is at the expense of EGFP and EBFP levels (this was already mentioned in the text).
- P2A and T2A constructs: there is a decrease in EGFP but a marked increase in EBFP. In these two cases, mKATE expression increases, making them interesting, as the reviewer notes.

This experiment provides a much-needed characterisation of parts often used in cell engineering experiments to achieve multi-gene expression. We provide insights that can support the design of multi-gene constructs with minimised resource footprint.

5. The TET-ON design exploration is interesting and could be impactful, given how widely used this inducible system is.

We want to thank the reviewer for the positive comment and for acknowledging that the TET-ON application is interesting and impactful.

However, the results and interpretation of the results need to be clarified. For example, it appears that the optimized resource-aware (multi-cistronic) designs, while driving significantly lower induction output levels relative to the canonical 2 TU design, lead to lower leakage which manifests as an improvement in fold change levels, at least in HEK293T cells. This represents an improvement in at least one gene expression dimension (fold change). Given these trade-offs, the authors should be clear about what they mean by maximized performance through resource-aware design optimization.

We thank the reviewer for the comment. We note that for the specific example pointed out by the reviewer, we used the expression “better performing circuits for gene expression control” rather than maximised performance. However, we agree with the reviewer that this can be better clarified, and we have thus modified the sentence at line 234:

“Our redesigned TET-ON serves as a proof-of-concept that genetic circuits for gene expression control can be designed to impose a lower demand on cellular resources while at the same time displaying improved fold change. Future work could focus on developing circuits with higher maximum output if desired for the application”.

Interestingly, this result does not appear to be consistent in CHO-K1 cells. The authors speculate “that the improved system may not reach full induction with the usual dox concentration, and that increased concentrations may be needed to improve the system dynamics in CHO-K1.” The authors should really provide evidence for this. Another interpretation is that they are getting lower rtTA expressed from the new designs.

We thank the reviewer for the question that allows us to characterise our TET-ON designs further. Following the reviewer’s suggestion, we investigated if higher dox induction would lead to higher expression levels from the system in both cell lines. However, dox concentrations up to 10ng/μl did not change the construct output in HEK293T and CHO-K1 (displayed below). We have included this experiment as a figure in the Supplementary Information (Fig. S13a).

We thus considered the second suggestion from the reviewer. We transfected our cell lines with the TET-ON constructs and an additional plasmid expressing the rtTA transactivator from the EF1a promoter. Adding rtTA did not increase EGFP expression and reduced capacity, as expected by the extra expression of the rtTA cassette (displayed below). Nevertheless, we noticed that the percentage of activated cells was higher in the presence of additional rtTA for both the IRES2 and P2A designs in both cell lines. We conclude that trade-offs need to be identified between the desired output and capacity. We have included this experiment as a figure in the Supplementary Information (Fig. S13b).

Fig. S13. Induction of three selected TET-ON designs. (a) Effect of increasing concentration of doxycycline on three selected TET-ON designs in HEK293T (left panels) and CHO-K1 (right panels). (b) Impact of additional rtTA transactivator on three selected TET-ON designs in HEK293T (top panels) and CHO-K1 (bottom panels). (c) EGFP fluorescence histograms for three selected TET-ON designs with or without co-transfection of a rtTA transactivator-expressing plasmid in HEK293T (top panels) and CHO-K1 (bottom panels). Test plasmid and capacity monitor expression levels are reported as mean RFU \pm std. Number of biological repeats for each sample is reported in Table S3. Data were analysed as described in the Methods section and Supplementary Note 1.

We have now added a new paragraph in the manuscript that discusses the new data (line 230):

“Increasing dox and rtTA concentration did not increase EGFP expression for the resource-aware designs (Fig S13a-b). However, we did observe that the percentage of EGFP-expressing cells was higher when additional rtTA was present for both IRES2 and P2A designs in both cell lines (Figure S13c). These findings suggest a balance between the desired output and capacity based on the application must be established.”

6. With respect to the results of Fig. 4b: only three samples are taken to draw correlation between EGFP and EBNA1. In order to draw a reliable correlation, the authors should choose more points to show whether there is reliable coincidence.

We thank the reviewer for the request because it uncovered that the correlation does not hold over a wider sample size. Please find below the new figure with EGFP and EBNA1 correlation (now Fig. S14a, left panel), correlation of mKATE when competing with EGFP vs mKATE when competing with EBNA1 (Fig. S14a, right), and western blot image (Fig. S14b). No correlation between EGFP and EBNA1 levels was now observed (Fig. S14a, left panel), which agrees with previous work reporting that functional modularity is affected by the genetic context (Shakiba et al. Cell Systems 2021). However, this does not diminish the importance of our framework. In Fig.

S14a (right panel), a good correlation between the mKATE levels in competition with EGFP and in competition with EBNA1 is shown, proving how the monitor output is still a good predictor of the resource limitation imposed by different circuit designs independently of the gene of use. We have included this figure in the Supplementary Information (Fig. S14).

Fig. S14. Output and capacity of EBNA1 designs in HEK293T. (a) Correlation between EGFP expression (RFU) and EBNA1 expression (normalised signal intensity) for the selected designs in Fig. 4 in HEK293T (left panel). Correlation between mKATE expression in competition with EGFP (RFU) and mKATE expression in competition with EBNA1 (RFU) for the selected designs in Fig. 4b in HEK293T (right panel). (b) EBNA1 (test plasmid) and Vinculin (internal control) expression for the selected EBNA1-expressing designs detected by Western blot. Capacity monitor expression is reported as mean RFU \pm std. EBNA1 expression is reported as normalised signal intensity and was calculated as described in the Methods section. Number of biological repeats for each sample is reported in Table S3. Data were analysed as described in the Methods section and in Supplementary Note 1.

We have now updated Fig. 4b (reported below) with the new data for both EBNA1 quantification and mKATE expression, providing further evidence that mKATE levels are conserved for the selected designs.

Fig. 4. Resource-aware construct design is useful for optimising genetic circuit performance and co-expression systems. (a) Minimisation of the resource load of the TET-ON inducible system by use of an IRES2 sequence. EGFP (left), mKATE (middle) and fold change (ratio of EGFP in the presence and absence of doxycycline induction) (right) are shown. **(b)** Resource-aware design of EBNA1-expressing constructs in HEK293T. Four designs with different resource footprints and similar EGFP expression are selected (diamonds, top left panel). EBNA1 (top middle panel) and capacity monitor (top right panel) expression levels for each design are shown. Four designs with different EGFP expression levels and similar resource footprints are selected (diamonds, bottom left panel). EBNA1 (top middle panel) and capacity monitor (top right panel) expression levels for each design are shown. EGFP and capacity monitor expression are reported as mean RFU ± std. EBNA1 expression is reported as normalised signal intensity and was calculated as described in the Methods section. Number of biological repeats for each sample is reported in Table S3. Mann-Whitney test P value: ****<0.0001, ***<0.0005, **<0.005, *<0.05. Data were analysed as described in the Methods section and Supplementary Note 1.

We have also updated our result section in the manuscript (line 246):

“While we observed an impact of genetic context on the test plasmid output, in accordance with previous reports (Shakiba et al. Cell Systems 2021), the expression levels of the capacity monitor did show correlation when in competition with either EGFP or EBNA1 (Fig. S14a-b). This indicates that the capacity monitor output is a good predictor of the resource footprint of a specific design independently of the gene of use. Five of the eight shortlisted designs were selected to yield similar EBNA1 output while displaying different resource footprints (Fig. 4b, top panels). In accordance with our previous characterisation (Fig. 3a-d), designs carrying strong promoters resulted in a higher resource footprint than designs with weaker promoters, with 13% remaining capacity for TETp, 53% for CMVp, and 68-85% for UBp and hACTBp designs (Fig. 4b, top panels). This is in line with the key findings we inferred from Fig. 3a-c. In addition, four designs bearing CMVp-Kz1 and one of four polyA configurations were selected to yield different EBNA1 outputs while maintaining the same resource footprint (Fig. 4b, bottom panels). In line with previous findings (Fig. 2c, Fig. S5), changing the polyA to more favorable configurations allowed for higher EBNA1 production without increasing the resource footprint.”

Moreover, the interpretation that "the higher-than-expected EBNA1 output from the UBp construct shows how dramatic co-optimisation of the output of both competing cassettes can be achieved by careful selection of genetic parts" seems unusual. Wasn't the result out of expectation or not coincident with their framework?

Following the reviewer's comment, we have now removed the sentence.

Minor comments and critiques:

1. *The authors should clarify what TETp design is used in Fig. 2a. I assume this is a constitutive version of the promoter?*

In the caption of Fig. 2a, we wrote, “Resource load of seven constitutive promoters and one inducible promoter”. We appreciate that this may not be clear, and we have thus clarified in the text that the TETp shown in Fig. 2a is an inducible promoter (line 77).

2. *Did the author normalize the number of each plasmid when transfecting cells? If so, was it done by molar ratio or by weight? This should be indicated either in the text or in the methods.*

This has now been added (line 323)

3. *In Fig. 3c, in the configuration of UBp-HGHpA, there is no expression of even in the capacity monitor plasmid. Can the authors explain?*

We thank the reviewer and apologise for not explaining this at first. We believe the reviewer refers to Fig. 2c. That configuration did not clone correctly, and after many attempts, we left it out of the library. We have now explained this in the figure caption.

4. *In the Capacity Table at the right of Fig. 3d, I recommend changing the listed order to be consistent with the plots on the left.*

We thank the reviewer for noticing this. We have now modified the figure accordingly (now Fig. 3f).

5. *Please clarify what cell line was used for Fig. 4 and Fig. S10-11? HEK293T or CHO-K1?*

We thank the reviewer for noticing that the info was missing. We have added this to the captions of Fig. 4 and Fig. S14.

Reviewer #2 (Remarks to the Author):

In this study, the authors characterized the role of resource competition in crosstalk between different cargos in mammalian cells, especially HEK293 and CHO cells. By testing various promoters and other element combinations, the authors found a negative correlation between two output proteins. However, the finding is obvious, and the underlying mechanism and exploration are not supportive. Such competition has already explored in mammalian synthetic biology area (pmc4364186).

We want to thank the reviewer for taking the time to consider our manuscript. We considered the paper mentioned by the reviewer, “Model-guided quantitative analysis of microRNA-mediated regulation on competing endogenous RNAs using a synthetic gene circuit”. In the article, the authors present a mathematical analysis to assess the behaviour of miRNA-mediated competing endogenous RNA regulation. The research is based on the knowledge that miRNAs can regulate a variety of endogenous target RNAs and that binding to these targets is competitive. While we understand why the reviewer is pointing to this paper, highlighting that miRNA binding is competitive, we disagree that this makes our manuscript and findings obvious.

Previous works recently published in Nature Communications (Jones et al Nat Com 2020 and Qin et al Nat Com 2023) support the relevance of our work, showing how resource competition impacts construct behaviour. However, our work goes a step further, providing the first evidence of the role of essential genetic parts in driving resource competition. We clearly demonstrate that promoter choice dramatically impacts the expression of a competing cassette and to a much higher extent than Kozak sequences or polyAs in mammalian cells. This is strikingly different from what has been reported in bacteria (Ceroni et al. Nat Methods 2015) and, therefore, non-obvious. In addition, no previous report highlighted the role of polyA in shaping competition. We are also the first to characterise the resource footprint of genetic parts used for multi-gene expression in mammalian cells. Finally, we provide relevant applications to highlight the relevance of our work for the cell engineering and bioproduction community. Our manuscript builds on previous efforts published in top-tier journals (Frei et

al, Jones et al). While these have addressed resource competition by implementing complex circuit topologies (i.e. iFFLs), we propose an alternative approach devoid of complex engineering and the shortcoming of such architectures (such as output suppression of the gene of interest). By carefully selecting essential gene parts, we provide design rules for resource-decoupled expression in mammalian cells. Our work stands as the most comprehensive and up-to-date report providing design rules to diminish the resource footprint of synthetic constructs without compromising on the output. It is novel, timely, and a good fit for this journal.

While almost all the assays were performed by transfection, each expression cassette's intensity is regulated by promoter intensity and the transfection dose. To reach a balance between multiple cassettes, the intensity would be simply optimized by testing different doses of plasmids. Whether such competition existed in transfection with high plasmid dose is unclear.

We want to thank the reviewer for allowing us to strengthen our findings. First, we would like to point out that the plasmid dosage transfected in these experiments is already relatively low. In particular, it is less than half of what is recommended in the manufacturer's protocol and 10-50% of what is used in similar published work (Frei et al. 2020, Jones et al. 2020, Qin et al 2023, Nat Comm). Furthermore, a previous report by *Frei et al* showed that resource competition is still present at lower plasmid dosages in titration experiments.

Nevertheless, we performed a subset of our experiments with lower plasmid dosage (figure below). We observed clear competition between the co-expressed genes even for the lower tested plasmid dose, proving that the observed effects are not due to transfection of high plasmid doses. We have included this experiment as a figure in the Supplementary Information (Fig. S3a).

We have now added a new paragraph in the manuscript that discusses the new data (line 70):

“Reducing the dosage of transfected plasmids did not rescue resource competition (Figure S3a)”

Fig. S3. Decreasing gene dosage does not rescue resource competition. (a) Resource competition between co-expressed genes when different test and monitor plasmids dosages are co-transfected in HEK293T.

Major concerns:

1. *Almost all the assays were performed by transfection. The intensity of each expression cassette is not only regulated by the promoter intensity but also by the transfection dose. To reach a balance between several plasmids, the dose can be optimized to give rise to more precise control.*

Similar to the point above, based on our data and previous literature (Frei et al), it appears clear that tuning plasmid dose is not sufficient to rescue competition between co-expressed cassettes. Moreover, low plasmid levels would introduce biases and problems in detection due to low transfection efficiencies.

2. *Since the transfection in HEK293 cells can deliver a much high level of DNA into the cells, which will occupy a large portion of the resource. However, whether this phenomenon would exist is unclear when researchers utilize other delivery approaches, including AAV, Lentivirus, or mRNA delivery. Additionally, whether the competition only existed in transfection with much high level of plasmids is unclear.*

We thank the reviewer for the comments. We agree that understanding if competition only occurs at the transient level or also at the integrated level is relevant. To answer the comments, we adopted a HEK293T cell line with a single copy AAVS1 integration of a dox-inducible BFP integrated. We transfected the cell line with a construct bearing a CMV-mKATE expression cassette (figure below). At the moment of transfection, dox was added to the culture medium to induce BFP expression from the genome. When the BFP is genomically expressed, a decrease in mKATE expression from the transiently expressed mKATE plasmid is observed. This ensures that (i) competition occurs also following the expression of a gene present in a single copy, and (ii) competition can be observed between integrated and transfected systems. Further investigation on competition at the genomic level is ongoing in our lab and will, in the future, provide further information on resource competition in mammalian cells. We have included this experiment as a figure in the Supplementary Information (Fig. S3b).

We have now added a new paragraph in the manuscript that discusses the new data (line 70):

“Reducing the dosage of transfected plasmids did not rescue resource competition (Figure S3a), and induction of a single-copy integrated gene led to resource loading on the capacity monitor (Figure S3b), indicating that resource competition is not limited to co-transfected systems”.

Fig. S3. Decreasing gene dosage does not rescue resource competition. (b) Resource competition between a single-copy integrated (dox-inducible BFP) and a transient payload (constitutive mKATE) in HEK293T. Test plasmid and capacity monitor expression levels are reported as mean RFU \pm std. Number of biological repeats for each sample is reported in Table S3. Data were analysed as described in the Methods section and Supplementary Note 1.

3. Authors contributed the underlying mechanism to transcriptional regulation rather than translation regulation due to the apparent relationship found in the promoters chosen but not the Kozak sites. The Kozak sequence is not a clear translational start signal. The strong transcription by powerful promoters such as CMV would affect all the resource allocation. Deep exploration to test the critical resources, including ribosome, polymerase, transcriptional factors, or tRNA, would be helpful for further research.

We thank the reviewer. We point to the answer provided to reviewer 1 in point 2 on a similar comment.

We note that in our experiments, Kz1 and Kz2 have very different output levels in all the tested promoter configurations and, therefore, reliably enable different translational efficiency.

We also tested three Kozak configurations for each promoter in the library, and weak promoters did not show any signs of competition for translational resources. Interestingly, in the presence of strong promoters, we noticed an impact on the capacity monitor, probably due to increased mRNA production and saturation of translational resources. This is in contrast with the hypothesis of the reviewer.

Finally, our paper presents important novel information. From our experiments, translational resources only appear to drive competition in particular configurations and only to a minor extent. From this, it can be concluded that translational resources acting downstream of the Kozak (i.e. tRNA, aminoacids, aminoacyl-tRNA synthetase, etc.), some of which are mentioned in the reviewer's comment, are not limiting in the tested cell lines.

4. The authors did not provide applications that face the competition problem. It would be great if the authors could show the applications to reveal their importance.

We thank the reviewer for the comment, but we disagree. We provide two examples that clearly face the problem of resource competition.

First, we optimised the design of a TET-ON system to reduce its demand for cellular resources. We showed that the original 2TU configuration strongly impacts the expression of a competing cassette and that, by using IRES or 2A elements, such impact can be dramatically reduced. As acknowledged by reviewer 1, resource-aware optimisation of the TET-ON system is interesting and impactful per se, given how often this system is used in mammalian cell engineering. We believe that our proof-of-concept experiment is not only useful for its intended application but also relevant to anyone seeking to optimise the expression of multi-gene constructs for any other purpose.

Secondly, we demonstrated resource-aware design optimisation of co-expression systems. We used EBNA1 and mKATE for our proof-of-concept experiment. We chose EBNA1 because it is a protein widely used in bioprocessing and biotherapy applications. We decided to keep mKATE to have an easy way to assess the resource footprint of the EBNA1 designs. Nevertheless, we stress that the mKATE could be replaced by any protein of therapeutic value. By co-transfecting EBNA1 and mKATE, they will inevitably compete for resources; therefore, this application faces the resource competition problem. Informed by our previous characterisation, we optimised the EBNA1-mKATE co-expression system by acting on the design of the EBNA1-expressing constructs. To be specific, our designs yielded: i) increased EBNA1 levels while maintaining a fixed resource footprint, ii) decreased resource footprint while maintaining a fixed EBNA1 expression, and iii) both increased EBNA1 expression and decreased resource footprint. Our proof-of-concept experiments represent a novel approach that can be useful to anyone seeking to optimise co-expression systems in mammalian cells.

Finally, papers recently published in Nature Communications provided proof-of-concept applications similar to ours to support their findings (Frei et al. 2020, Jones et al. 2020). This reiterates the suitability of our applications.

References

Ceroni, F., Algar, R., Stan, G. B. & Ellis, T. Quantifying cellular capacity identifies gene expression designs with reduced burden. *Nat Methods* **12**, 415+, doi:10.1038/Nmeth.3339 (2015).

Eisenhut, P. *et al.* Systematic use of synthetic 5'-UTR RNA structures to tune protein translation improves yield and quality of complex proteins in mammalian cell factories. *Nucleic Acids Res* **48**, e119, doi:10.1093/nar/gkaa847 (2020).

Frei, T. *et al.* Characterization and mitigation of gene expression burden in mammalian cells. *Nat Commun* **11**, 4641, doi:10.1038/s41467-020-18392-x (2020).

Jones, R. D. *et al.* An endoribonuclease-based feedforward controller for decoupling resource-limited genetic modules in mammalian cells. *Nat Commun* **11**, 5690, doi:10.1038/s41467-020-19126-9 (2020).

Qin, C. *et al.* Precise programming of multigene expression stoichiometry in mammalian cells by a modular and programmable transcriptional system. *Nat Commun* **14**, 1500, doi:10.1038/s41467-023-37244-y (2023).

Shakiba et al. Context-aware synthetic biology by controller design: Engineering the mammalian cell. *Cell Systems* **12**, 6, 561-592 doi.org/10.1016/j.cels.2021.05.011 (2021).

Reviewers' Comments:

Reviewer #1:

Remarks to the Author:

The authors have satisfactorily addressed the necessary comments, resulting in a clearer and improved manuscript.

However, I do have one question regarding TETp, which is referred to as an inducible promoter. In Fig2a (left), TETp (assuming this is without DOX) shows the strongest EGFP expression, even higher than EF1ap. Do you think this is a reasonable result? Apart from this, I have no additional critiques. As a minor comment, I noticed that the colors in Fig3b (right) should be changed to match the legend (e.g., the legend shows purple, but the figures show gray).

Reviewer #2:

Remarks to the Author:

Thank the authors for addressing the concerns raised during the review process. Unfortunately, I cannot recommend acceptance for publication in Nature Communications as the authors have not adequately addressed the raised concerns.

Firstly, while the study of resource competition in mammalian cells is an important topic, the most data presented were limited to fluorescence protein expression and plasmid transfection in HEK293 cells, failing to convincingly demonstrate the phenomenon's wider application in mammalian cells beyond what is already known.

Furthermore, the manuscript lacks an adequate demonstration of potential factors other than promoter availability that may affect resource competition within cells.

Therefore, I suggest a more specialized journal for consideration.

Point-by-point responses to reviewer comments

Reviewer #1

Remarks to authors: The authors have satisfactorily addressed the necessary comments, resulting in a clearer and improved manuscript.

We thank the reviewer for the positive comments on our revised manuscript.

However, I do have one question regarding TETp, which is referred to as an inducible promoter. In Fig2a (left), TETp (assuming this is without DOX) shows the strongest EGFP expression, even higher than EF1ap. Do you think this is a reasonable result? Apart from this, I have no additional critiques.

We thank the reviewer for the question. The TETp data shown in Figure 2a represent the EGFP output of the TETp construct when maximum dox concentration of 1ng/ μ l is added. We thank the reviewer for noticing this. We have now specified in the caption of Figure 2 that the construct is induced.

As a minor comment, I noticed that the colors in Fig3b (right) should be changed to match the legend (e.g., the legend shows purple, but the figures show gray).

We thank the reviewer for noticing this. We have now modified Figure 3b accordingly.

Reviewer #2 (Remarks to the Author)

Thank the authors for addressing the concerns raised during the review process. Unfortunately, I cannot recommend acceptance for publication in Nature Communications as the authors have not adequately addressed the raised concerns. Firstly, while the study of resource competition in mammalian cells is an important topic, the most data presented were limited to fluorescence protein expression and plasmid transfection in HEK293 cells, failing to convincingly demonstrate the phenomenon's wider application in mammalian cells beyond what is already known. Furthermore, the manuscript lacks an adequate demonstration of potential factors other than promoter availability that may affect resource competition within cells.

We thank the reviewer for the comment. However, we do not fully understand the concerns of the reviewer. The majority of the data shown in the manuscript are relative to two cell lines, HEK293T and CHO-K1. This has allowed us to underline the difference in behaviour and competition profiles of the presented constructs when different genetic backgrounds are adopted. We highlighted this, for example, in the case of polyA contribution to resource competition. We agree with the reviewer that future work will be needed to investigate the same in more cell lines and further compare cell-line specific effects.

Regarding the sole use of fluorescent reporters, we would like to point to Figure 4 where we introduced a second protein, in place of EGFP, in the test constructs (i.e. EBNA1). This was done on purpose, to extend considerations of resource competition to different proteins, not restricted to fluorescent reporters.

Finally, while it is true that we point to promoter strength as an important factor impacting the resource footprint of a construct, we also prove that Kozak sequences are not as important in shaping competition. This gives key insights on the relative importance on transcriptional and translational resources in driving competition. This was not appropriately demonstrated before, as shown by contrasting evidence presented in the literature. Together with this, we also clearly introduce polyA sequences as strong determinants of the resource competition response when different cassettes are co-expressed in mammalian cells. To the best of our knowledge, this was not previously shown. Finally, we characterise the case of multigene constructs and their syntax in respect to resource competition, showing how this can impact the performance of a test biosensor. Overall, the importance of these novel findings should be apparent when looking at our applications. However, we agree with the reviewer that further work will be necessary in the future to add further characterisation to our results, and we have now provided considerations on this in a new paragraph added to our Discussion (line 293):

“Future work investigating these effects in a wider panel of cell lines could provide a comprehensive analysis of cell line-specific effects. While we report initial evidence that competition arises also between transient and integrated systems, further work will also be needed to expand on this and provide characterisation and analysis of how our findings translate to integrated systems. This will be of invaluable importance considering the wide adoption of stably integrated constructs in mammalian cell engineering and its applications”.